# COMplex mental health PAThways (COMPAT) Study: A mixed methods study to inform an evidence-based service delivery model for people with complex needs: Study protocol

**Pooja Saini** [1]*, **Antony Martin** [2], **Jason McIntyre** [1], **Anna Balmer** [1], **Sam Burton** [1], **Hana Roks** [1], **Laura Sambrook** [1], **Amrith Shetty** [3], **Rajan Nathan** [1,3]

1 Faculty of Health, Liverpool John Moores University, Liverpool, United Kingdom, 2 QC Medica, York, North Yorkshire, United Kingdom, 3 Cheshire and Wirral Partnership NHS Foundation Trust, Chester, United Kingdom

* p.saini@ljmu.ac.uk

**Funding:** PS received the award for this study. Grant number: 1582 CWP. The study is funded by

## Abstract

### Background

Mental health services for adults, as they are currently configured, have been designed to provide predominantly community-based interventions. It has long been recognised that some patients have such significant clinical and/or risk needs that those needs cannot be adequately met within standard service delivery models, resulting in a pressing need to consider the best models for this group of people. This paper shares a protocol for a mixed methods study that aims to understand: the profile and history of service users described as having complex needs; the decision-making processes by clinicians that lead to complex needs categorisation; service users and carers experience of service use; and, associated economic impact. This protocol describes a comprehensive evaluation that aims to inform an evidence-based service delivery model for people with complex needs.

### Methods

We will use a mixed methods design, combining quantitative and qualitative methods using in-depth descriptive and inferential analysis of patient records, written medical notes and in-depth interviews with service users, carers, and clinicians. The study will include five components: (1) a quantitative description and analysis of the demographic clinical characteristics of the patient group; (2) an economic evaluation of alternative patient pathways; (3) semi-structured interviews about service user and carer experiences; (4) using data from components 1–3 to co-produce vignettes jointly with relevant stakeholders involved in the care of service users with complex mental health needs; and, (5) semi-structured interviews about clinical decision-making by clinicians in relation to this patient group, using the vignettes as example case studies.

### Discussion

The study's key outcomes will be to: examine the resource use and cost-impact associated with alternative care pathways to the NHS and other sectors of the economy (including

the Cheshire and Wirral Partnership NHS Foundation Trust. https://www.cwp.nhs.uk/ Prof Rajan Nathan is the Clinical Lead on the project, AS is the Medical Director and both have had a role in study design, decision to publish and commenting on the manuscript.

**Competing interests:** The authors have declared that no competing interests exist. The study was commissioned by Wirral Clinical Commissioning Group and routed through Cheshire and Wirral Partnership NHS Foundation Trust (CWP) to Liverpool John Moores University. TN and AS are employed by CWP and work for the health organisation where the study is taking place. No other relationships or activities could appear to have influenced the submitted work. The funding commissioners had no role in the design of the study; in the collection, analyses, or interpretation of data; or in the writing of the manuscript.

social care); explore patient health and non-health outcomes associated with alternative care pathways; and, gain an understanding of a complex service user group and how treatment decisions are made to inform consistent and person-centred future service delivery.

## Introduction

Recent recommendations for effective support from mental health services suggest that individuals presenting with complex behavioural and mental health needs are less likely to receive the provision of care they require due to their need for longer-term, highly specialised support [1]. Complex mental health needs may include some people who have such significant clinical and/or risk needs that those needs cannot be adequately met within the generic mental health services. The majority of these people have a diagnosis of psychosis, severe negative symptoms, and cognitive impairments. Many also have coexisting mental health problems and physical health concerns resulting from poor lifestyle conditions and side effects of psychotropic medication [2]. Thus, there is a pressing need to inform an evidence-based service delivery model for mental health service users with complex needs.

Studies examining the profile of this group indicate that it is not just a matter of the extent of the need, but also the complexity of their clinical profile and history [2,3]. Whilst complexity appears to be a key factor common to these people [1], there is little understanding of why a person becomes identified as someone in this group. A delineation of the components of 'complexity' in this context will not only provide an evidence base to support the development of appropriate services, but also facilitate a 'prevention' approach in which the model of assessment and intervention at an earlier point may reduce the likelihood of the person becoming 'complex'.

Mental health services for adults, as they are currently configured, have been designed to provide predominantly community-based interventions. These community services are supplemented by additional provision that is accessed on the basis of acuity/risk (i.e. inpatient services) or of diagnostic specificity. Individuals presenting with complex needs are often accommodated in out-of-area placements that are a long distance from their loved ones and communities [4], due to the inability, or arguably the unwillingness [3], of local services to meet their needs. There are growing concerns about the impact of out–of-area placements on mental health service users, both clinically and financially [5]. In addition to being costly to the NHS and local social care authorities, individuals placed out-of-area can become socially dislocated, achieve poorer outcomes [6], experience disruptions to their lives [7] and in some cases, be over-supported [5]. The issue of distance can also cause complications for the 'home' services who made the referral, which are services generally provided in the locality of a patient's home, as it can be difficult to maintain contact regarding the suitability of the placement and the person's care, which can also hinder their rehabilitation and eventual reintegration into their home community [4,8].

Within this exploration of an effective model for service delivery, it is also beneficial to explore the experiences of carers. It has been highlighted that caring for someone with a mental health condition can have a strain on the carer themselves, with physical health and mental health impacts, including anxiety and depression, being associated with informal care [9]. Currently, there is limited literature available which focuses on the experiences and feelings of carers of inpatients who are placed in out-of-area psychiatric placements. Previous studies have noted carers experienced difficulties with access to information, mainly due to confidentiality

issues, as the person had denied access of their carer to their information [10,11]. A literature review by Askey et al. [11] found that carers of inpatients with complex needs struggled to communicate with staff and felt that they were excluded by the service team and their views were regarded as of little importance. Wilkinson and McAndrew [12] found that carers had felt powerless and expressed a need to be valued and recognised by service providers.

Although previous research has been conducted about this group of people in other areas of the country, there are limitations on the local applicability of this data. Furthermore, other studies have focused exclusively on the person profile and have ignored a major contributor both to the identification of these people as in need of 'specialist' placements and the wide variability in the way different clinicians and services respond to people.

Research has found that clinician decision-making is not based on an objective analysis of person characteristics [13–15]. Rather, decisions are influenced by subjective human factors and oft-ignored systemic dynamics; predicting, managing and responding to pressures on the inpatient resource require greater clarity about the factors that inform decision-making in mental health settings [16]. Decisions regarding placement arrangement and clinician response to individuals with complex mental health needs are not just a result of the appraisal of clinical and risk-related information; emotional, interpersonal, and contextual factors are also critical. Understanding these influences allows the development of a holistic model that does not just provide the best approach for this group, but also ensures consistency and predictability of approach by decision-makers.

Previous studies on decision-making in mental health settings have tended to concentrate on disorder-based and person-based factors [17–19]. There have been some studies examining specific aspects of clinical decision-making such as shared-decision making [20,21], the accuracy of decision-making [22], the role of human factors [23] and the influence of the way risk is framed [14]. Nevertheless, there has been limited empirical analysis of how decisions are made in practice. Lombardo and colleagues [24] examined decision-making by mental health crisis team clinicians; however, a review of the literature did not identify any study that had specifically examined the wide range of factors influencing mental health practitioners' decision-making in relation to acute hospital admissions. Gaining a more in-depth understanding of these factors would inform approaches to clinical training and supervision of clinicians. Furthermore, service delivery models should take account of the way decisions are made in practice.

This study aims to examine the resource use and cost impact associated with alternative care pathways to the NHS and other sectors of the economy (including social care) and explore patient health and non-health outcomes associated with alternative care pathways. The focus of our study is to gain an in-depth understanding of a cohort of service users with complex mental health needs and how decisions are made in their treatment to inform consistent and person-centred service delivery; particularly for people in out-of-area placements. Service users with complex mental health needs were recognised as a priority group who meet the criteria for this study due to their earlier-discussed needs that currently cannot be adequately met within the standard mental health service model.

## Study objectives and aims (Table 1)

This study involves a comprehensive evaluation of: (i) the profile and histories of people currently defined as having complex mental health needs; (ii) the decision-making processes that lead to these people entering this complex group (and by extension are liable to find themselves in specialist placements); and (iii) patients' experiences of service use.

Secondary aims include collating in-depth information about carers' experiences, (i.e. individuals who are caring for service users with complex mental health needs who may find

**Table 1. Study objectives and research questions.**

| Study Objectives | Specific research questions |
|---|---|
| Evaluate the profiles and histories of service users with complex mental health needs | What factors lead to a service user being defined as having complex mental health needs? |
| Evaluate the decision-making processes that lead to these service users entering this complex group | What influences the decisions made by clinicians that lead service users to become defined as complex? |
| Evaluate the experiences and views of service users who may find themselves in a specialist placement | What are the experiences of service users who are in a specialist placement? |
| Evaluate the experiences of carers, who care for a service with complex mental health needs who may find themselves in specialist placement | What are the experiences of carers who care for people with complex mental health needs? What are the experiences of carers who care for people in specialist (out of area) placements? |
| Evaluate clinician's experiences of treating individuals with complex mental health needs | What are clinicians' experiences of treating individuals with complex meant health needs? |

themselves in specialist placements) and, clinician's decision-making processes when treating individuals defined as having complex mental health needs.

## Methods

### Study setting

This study will be conducted in Cheshire and Wirral Partnership NHS Foundation Trust (CWP) based in Chester, a large UK-based NHS provider of community and hospital-based mental health services in Northwest England.

### Study design

This study will use a mixed methods approach. Quantitative data will be derived from an in-depth analysis of patient medical records. Semi-structured interviews with service users, carers and clinicians will be used to obtain qualitative data. For qualitative analyses, approximately 10 participants will be included in each of the three participant groups to ensure thematic saturation. The qualitative component of this study will allow for the exploration of the more complex aspects of care, such as patient experiences of services and decision-making, and as such contribute crucially and in a unique way to the quantitative data collection.

## Quantitative study

### Recruitment

To address aim (i), 272 patient data files will be extracted, with 80 invited for in-depth analysis. Service users in out of area placements will be identified using financial payment records and cross referencing these with clinical records held by staff monitoring the placements, to ensure all cases are identified. In respect of the suitability of the service users who were in out-of-area placements, two clinicians on the team will assess, based on a review of the service users' clinical records, who should be included for the study (see Fig 1).

Out of the 80 people, 40 will be service users who are in inpatient placements (e.g. super-stranded [hospitalised for over twenty-one days [25]], out of area placements and rehabilitation) and 40 who meet the criteria of being service users with complex mental health needs who are in community-based placements (e.g. home care treatment, supported accommodation). In terms of inclusion criteria, participants will be recognised by the Trust's clinicians as having complex and long-term recovery needs and longstanding mental health problems, who may have had out-of-area placements in the last five years. Participants will be excluded from

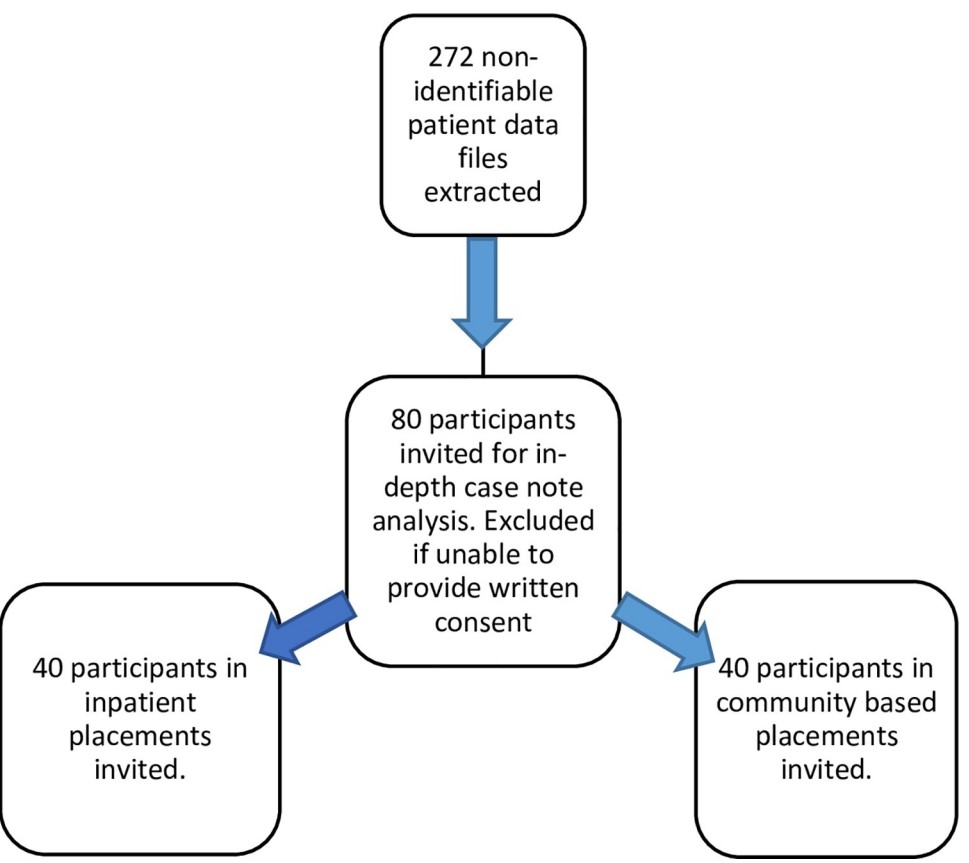

**Fig 1. Recruitment phases for the quantitative data.**

taking part in the study if they are under the age of 18. In respect of the 80 participants invited for in-depth analysis, they will be excluded if they are unable or unwilling to provide written informed consent to participate in the study. Post-hoc power calculations will be conducted for quantitative inferential analysis, as the quantity and quality of extracted data is unknown. Moreover, the novelty of the research prevents the formulation of expected effect sizes for power calculations.

## Design

A retrospective cohort design will be employed to assess patients' pathways to current placement, along with their demographics, clinical profiles, and risk profiles. Patient and Public Involvement (PPI) is extremely important to ensure the proposed research meets the needs of the target population. We have taken the approach described in Table 2 towards involving the public in the development of this study.

## Materials

A proforma was developed for extraction of data from clinical records, with input from relevant stakeholders comprising representatives of Cheshire and Wirral Partnership, patient engagement, the commissioners, the Local Authority, and housing. Metrics included were conventional records (e.g. demographics, diagnosis, placement), a wider range of data including that to undertake the economic analysis and data describing other relevant aspects of

**Table 2. Patient and Public Involvement.**

| Patient and Public Involvement (PPI) | | | |
|---|---|---|---|
| Insight work was undertaken with stakeholders, which informed the development of the study design including the recruitment procedures for participants into the study and the choice of data collection tools to be used within the study. | We are engaging with the NHS Trust PPI groups, including carers and other people working within community settings to become members of the Service User Advisory Group. | The public will be engaged at an early stage of the evaluation, and views and feedback will be considered at each stage of the project. Two members of the Service User Advisory Group will attend all steering group meetings and be involved at each stage of the research, from the design, analysis, interpretation of findings, dissemination, and further implementation work. | Two members of the steering board will attend the Service User Advisory Group in order to feed information from each group at all times. |

**Abbreviations:** NHS, National Health Service; PPI, Patient and Public Involvement.

patients' experience (e.g. developmental history, housing difficulties, mental health and social care pathways). The proforma will be piloted and any modifications will be made prior to data collection. For service users in Cheshire and Wirral Partnership NHS Foundation Trust, demographic, clinical and service utilisation data will be gathered in situ by experienced researchers (assistant/research psychologists) who will receive further training from onsite research and development managers.

## Data management

No identifiable patient data will be extracted to the proforma templates. Data will be transferred from the proforma templates to an Excel document and recoded for data analysis using SPSS and STATA software.

Participants contact details, audio interview files and verbal consent recordings for all aspects of the study will be stored on a secure, password protected database and backed up on the Liverpool John Moores University (LJMU) server.

## Data analysis

Descriptive analysis will be conducted to produce a clinical and demographic profile of the patient group. Inferential statistical models, such as linear regression, logistical regression, and ANOVA, will be conducted to compare clinical outcomes between different treatment pathways and to identify predictors of better clinical outcomes for patients while adjusting models for potential confounding variables.

## Qualitative studies

### Recruitment

**A) Service users.** To address aim (iii), 10 service users with complex needs will be interviewed about their experience of contact with relevant services and of the way decisions about their care were made. In terms of inclusion criteria, participants will be individuals recognised by the Trust's clinicians as having complex and long-term recovery needs and long-standing mental health problems, who may have had out-of-area placements in the last five years. Participants will be excluded if they are under the age of 18 or are unable/unwilling to provide written informed consent to participate in the study (see Fig 2A).

**B) Carers.** To address our secondary aim, ten carers of individuals with complex needs will be interviewed. In terms of inclusion criteria, participants will be carers of service users with complex mental health needs recruited via the Trust's clinicians. Participants will be

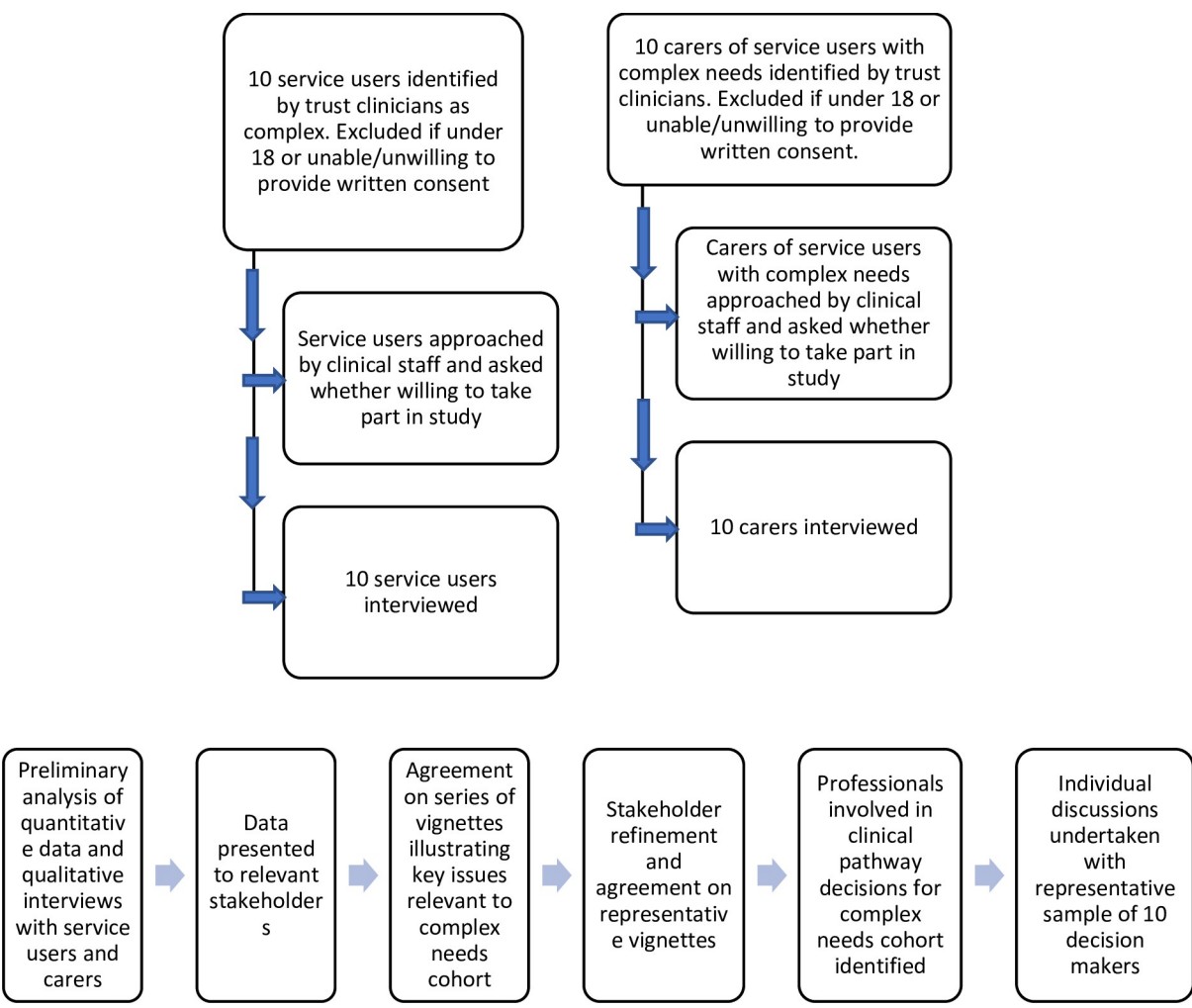

**Fig 2.** A. Recruitment phases for the qualitative interviews with service users and carers. B. Recruitment phases for the development of vignettes for the qualitative interviews with clinicians.

excluded if they are under the age of 18 or are unable/unwilling to provide written informed consent to participate in the study (see Fig 2A).

**C) Stakeholder workshops.** Following preliminary analysis of the quantitative data and qualitative interviews with service users and carers, the data will be presented to a group of relevant stakeholders (see Fig 2B), comprising representatives of Cheshire and Wirral Partnership, patient engagement, the commissioners, the Local Authority and housing, in order to agree a series of vignettes illustrating the key issues relevant to service development for this cohort of patients. Over three meetings, the stakeholder group will refine and agree a small number of representative vignettes, to be used within the clinician interviews (discussed below).

**D) Clinicians.** Professionals involved in making clinical and pathway decisions for this cohort will be identified and individual discussions will be undertaken with a representative sample of 10 decision-makers (see Fig 2B). This will address aim (ii) and one of our secondary aims.

## Design

This aspect of the study will employ a qualitative approach, using semi-structured interviewing to explore the experiences and views of individuals with complex mental health needs, as well

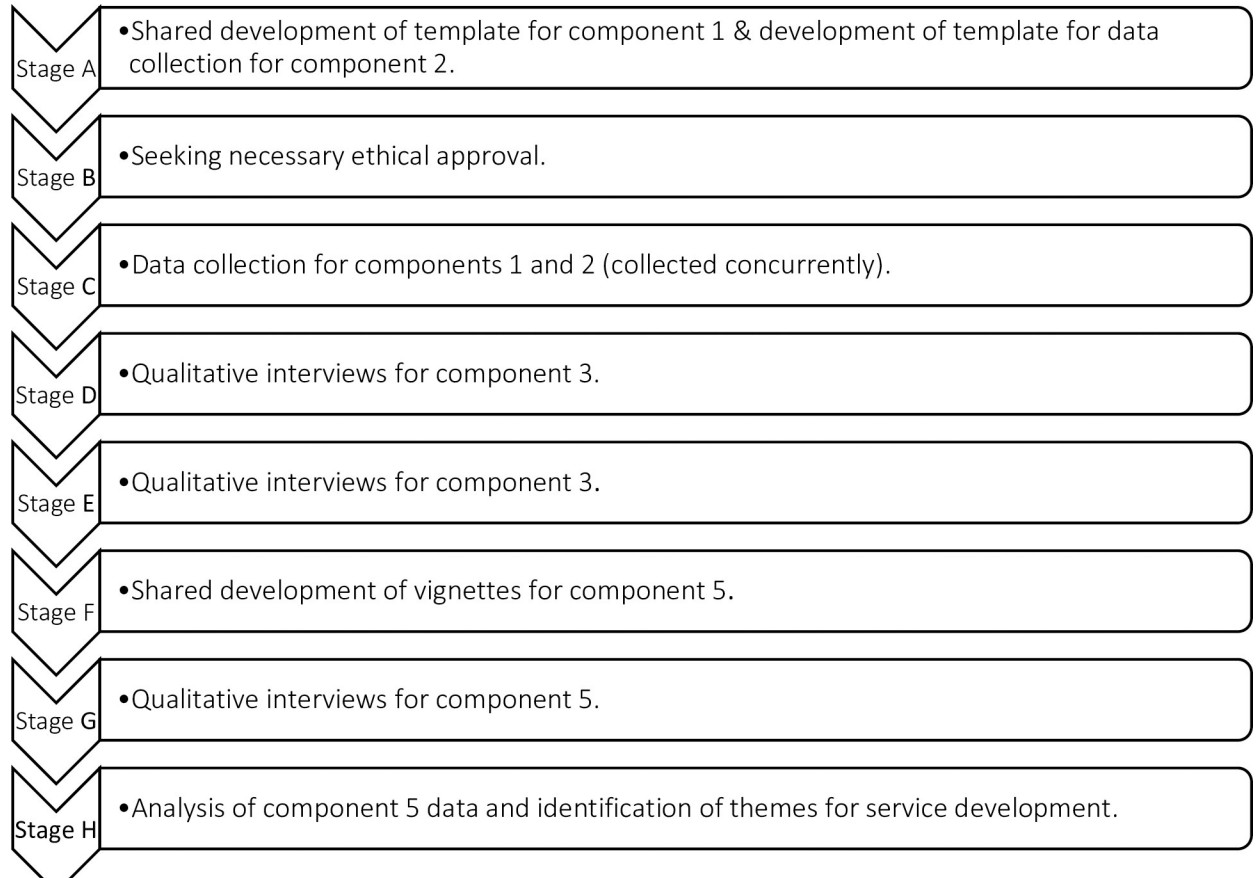

**Fig 3. Detailed study plan.**

as that of their carers and the clinicians involved in their care. People within this group from different types of placement and their carers will be approached to ask whether they are willing to speak to us about their experience of contact with relevant services and of the way decisions about their care were made. Participants will only be identified and approached by clinical staff who will use their clinical judgement to invite participants. The detailed study plan can be reviewed in Fig 3.

## Materials

Participants will be provided with a participant information sheet and consent form to sign/verbally consent prior to taking part in the research. Semi-structured interview schedules have been designed with questions to facilitate discussion about experiences of care and decision-making, with relevant prompts included to guide the discussion if necessary. The interviews will be recorded using an audio recording device.

## Procedure

Once written informed consent has been provided, the interviews will be undertaken remotely, due to the ongoing COVID-19 restrictions and to reduce disruption of carers' day-to-day work and caring responsibilities. It is anticipated that all interviews will last around 30–45 minutes. During the interviews, service users will be asked in general terms to discuss their

experiences of contact with mental health services, as well as the care they received. Prompts have been included to guide the discussion if necessary, covering areas such as involvement in decision-making, autonomy, placements, psychological therapies, relationships with staff and experience of discharge. Carers will be asked about their experiences of caring for someone with a complex mental health need, the admissions process and if there are any additional comments they would like to add.

Regarding the clinician interviews, the discussions will use the vignettes created during the stakeholder workshops to guide a discussion about (a) factors that contribute to the way decisions are currently made, (b) the support necessary to improve decision-making, and (c) alternative approaches for people who present with a similar clinical/risk profile. Semi-structured interview schedules have also been designed to facilitate further discussion around decision-making. During the interviews, clinicians will be asked about their experiences of treating service users with complex mental health needs and the decision-making processes that take place when managing these individuals.

## Data analysis

Discussions will be recorded, transcribed verbatim, and subjected to thematic analysis to identify important areas that a new service delivery model should attempt to address. The findings from the thematic analysis of clinician interviews will be used as a framework for the stakeholder group in order to agree the detail of improved service models for this group of people.

Transcripts will be checked against the audio files for accuracy. The analysis of all transcripts will be conducted and discussed by various members of the research team, each with different disciplinary backgrounds. The data will be analysed following the principles of qualitative thematic analysis using NVivo software [26], adopting a critical realist perspective and using the framework approach. Normalisation Process Theory (NPT) will be adopted as a broad framework through which to make sense of the qualitative data and draw conclusions relating to how readily the proforma might be implemented amongst other NHS Trusts and embedded into health care systems. Analysis will follow the five stages of framework analysis: familiarisation with the data; identifying a thematic framework; indexing the data; charting the data; and mapping and interpretation.

The iterative coding process will enable the continual revision of themes until the final classifications of major themes can be agreed by the team. During repeated rounds, frequent comparisons will be made across codes and the interview data to develop, review, and refine themes [26] on the basis of the complementarity, convergence, and dissonance of ideas across data sources [27]. To establish procedural reliability and conceptual credibility [28], additional members of the research team with experience in qualitative methods will examine a sample of transcripts to compare their perceptions of the interview data and analysis with the main analyst's interpretation. All findings will then be critically tested within the research group. Any disagreements will be resolved by discussion.

## Health economic analysis

An examination of resource use and cost-impact associated with alternative care pathways to the NHS and other sectors of the economy (including social care) will be conducted. A detailed patient-level cost analysis (bottom-up cost analysis) will be conducted through: (1) the identification of a relevant list of health resources consumed; and (2) quantification of resources in physical units (and time stamps for use), including direct medical costs and non-medical costs. Health care resource use data will primarily be extracted from patient records and

combined with unit costing data from Personal Social Services Research Unit (PSSRU) costings, NHS national reference costs, and published health economic and costing studies.

An exploration of patient outcomes associated with alternative care pathways will also be conducted. This analysis will further examine demographic covariates (including gender, age, postcode/LSOA) and health and non-health outcome data associated with alternative patient pathways.

## Ethical considerations

Ethical approval was obtained from the NHS Health Research Authority and West Midlands —Coventry & Warwickshire Research Ethics Committee: [REC Ref: 21/WM/0020] Integrated Research Application System (IRAS) prior to study commencement. Ethical approval was received on 19[th] March 2021 from HRA and Health and Care Research Wales (HCRW). The study will be undertaken in compliance with the research protocol. All participants will be given a participant information sheet and consent form prior to taking part in the research. Personal data will be documented in a password protected and encrypted computer.

### Data handling

All confidential data will be stored securely on the University research centre site with strictly limited access. Participants will be allocated an ID code which will be used on documents such as proformas to maintain confidentiality and minimise the use of personal data. The study Sponsor is Liverpool John Moores University who takes primary responsibility for ensuring that the design of the study meets appropriate standards in accordance with Good Clinical Practice (GCP) guidelines. All data will be handled according to the General Data Protection Regulation (GDPR) 2018. Confidential data will be stored securely on site with limited access.

### Safety monitoring

Adverse events and risk standardised operating procedures will be developed and will be followed by all researchers working on the study. Adverse events are defined as significant negative episodes, or significant deterioration in condition, which happen to participants during their participation in a study. These will be reported by research assistants to senior research staff and clinicians, who will ascertain whether these are thought to be linked to participation in the study. In terms of participant safety, attempts will be made to ensure that no explicitly sensitive questions are asked; however, there is always a possibility that participants may become distressed when discussing their experiences of mental health services, particularly if these were negative. There is a low risk of some participants finding discussions about difficult experiences or emotions uncomfortable or resulting in distress overall. A number of steps have been taken to minimise this risk. For example, all participants will be informed of the possible risk of distress in the participant information sheet. Risk of distress will also be discussed with participants verbally by the researcher before they are asked to consent to take part in an interview. A risk protocol, which was co-developed by experts-by-experience (with lived experience of mental health), clinicians, and researchers will be used to manage the potential risk of distress during the study. All participants will be informed they do not need to answer any questions they do not wish to and will be free to withdraw from the study at any time without detriment to themselves. Participants will be able to take a break should they become distressed during an assessment (possibly stopping altogether or taking a break depending on how the participant wants to proceed). All researchers will receive training in identifying and responding to distress by chartered psychologists involved in the research team. All participants will be provided with signposting information relating to local sources of help and support and will be encouraged to

make use of this list if they experience distress or emotional discomfort during or after the study. All participants will be under mental health services with a clinician, therefore if any participants did present with an increased level of risk during the study, the researchers would follow the risk protocol, which includes detailed procedures for escalating levels of risk.

## Discussion

This study aims to address questions that are of central relevance to the provision of specialist mental health services and as such the results will have the potential to immediately impact on the way such services are commissioned and delivered. An additional strength of the study is that not only does the design allow characterisation of the service users' histories and their experience of services, but it also includes a focus on how clinicians come to judgements about services users. The study key outcomes will be to: examine the resource use and cost impact associated with alternative care pathways to the NHS and other sectors of the economy (including social care); explore patient health and non-health outcomes associated with alternative care pathways; and gain an understanding of a complex service user group and how decisions are made in their treatment to inform how services are delivered in the future and made more person-centred and consistent.

## Strengths and limitations to the study design

Retrospective case note studies are commonly used within medical research [29,30]. Within medical research, retrospective case note research has examined both physical [29,30] and psychiatric conditions [31,32]. In psychiatric care, case note research has been shown to predict seclusion [32], at-risk groups for PICU referral [33], and the need for multi-faceted diagnostic tools to ensure accuracy [34].

Case notes are not just aides-memoirs for doctors but are complex documents that can be used for teaching, research, and clinical audit, as well as evidence in the event of litigation. Information obtained when a person is admitted informs the whole diagnostic and care planning process, including risk management strategies. It also follows that admission notes will point towards a diagnosis and impart a clear treatment plan to a greater or lesser extent. Psychiatric care medical records can also provide information about the management process of people in primary care, including consultation, treatment, and referral data. Previous research has highlighted methodological limitations in the use of medical record reviews including variations in accuracy or the amount of detail provided [35,36] and the risk of underestimating figures for consultations as not all are recorded in cases notes [37]. However, one study comparing psychiatric medical records and patient self-report questionnaires found similar figures for the mean number of consultations in both sources [35]. Additionally, a systematic review into the quality of computerised medical records revealed that the recording of consultations on such systems tended to be high [38,39].

In diagnostic assessment, perfect inter-rater reliability would occur when psychiatric practitioners could always arrive at the same diagnosis for a given patient. Inter-rater reliability and how practitioners come to decisions is often limited to assumptions drawn by them. Clinical involvement for accuracy is reliant on clinical records, which allow for the capture of behaviour prior to diagnoses, that may not be captured in typical records, allowing the identification of meaningful predictors for diagnoses [32,40].

## Dissemination

The study will have real-world impact via collaboration with clinicians, patients and public members, academics, and other relevant sector organisations. Study outcomes are relevant to

clinical practice and research, and will help improve understanding of the needs of this service user group. Our dissemination strategy is designed to maximise reach and impact of the results of this study across stakeholder groups.

For dissemination to be effective, dialogue is needed with relevant audiences. Project findings will be disseminated in close consultation with clinicians, community mental health professionals, public health, third sector organisations (e.g., housing associations), and those affected by complex mental health needs. The work will be of considerable interest to clinical and academic researchers in the field of mental health. Publications in high-impact journals, alongside presentations at regional, national, and international conferences will be pursued to maximise dissemination amongst academic and research audiences. A high specification executive summary of the key findings will be disseminated to clinical practitioners and researchers across the UK.

## Acknowledgments

The authors would like to thank Dr Rebecca Cummings and Dr Phil Elliott for their time and effort in support of the development of this study protocol. The authors would also like to thank the following stakeholders involved in meetings for the development of the study design and tools: Darren Birks (Wirral CCG), Robert Oxley (Wirral Local Authority), Lisa Newman (Wirral Local Authority–Housing), Angela Davies (Deputy Head of Service at CWP), Gagandeep Singh (Consultant Medic at CWP), Sean Boyle (Acute Care Director at CWP), Susie Williams (Team Manager), Bridget Hollingsworth (Wirral Local Authority), Sarah Batty (Rethink UK) and Peter Mudie (Public Advisor for the study).

## Author Contributions

**Conceptualization:** Pooja Saini, Amrith Shetty, Rajan Nathan.

**Funding acquisition:** Pooja Saini, Amrith Shetty, Rajan Nathan.

**Methodology:** Pooja Saini, Antony Martin, Jason McIntyre, Amrith Shetty, Rajan Nathan.

**Project administration:** Anna Balmer, Sam Burton, Hana Roks, Laura Sambrook.

**Supervision:** Pooja Saini, Rajan Nathan.

**Writing – original draft:** Pooja Saini, Anna Balmer, Sam Burton, Hana Roks, Laura Sambrook.

**Writing – review & editing:** Pooja Saini, Antony Martin, Jason McIntyre, Anna Balmer, Sam Burton, Hana Roks, Laura Sambrook, Amrith Shetty, Rajan Nathan.

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
