## [Decision Letter · Decision Letter 0]

7 Feb 2022

COMplex mental health PAThways (COMPAT) Study: a mixed methods study to inform an evidence-based service delivery model for people with complex needs: Study protocol

PONE-D-21-09749

Dear Dr. Saini,

We’re pleased to inform you that your manuscript has been judged scientifically suitable for publication and will be formally accepted for publication once it meets all outstanding technical requirements.

Kind regards,

Kathleen Finlayson

Academic Editor

PLOS ONE

Additional Editor Comments (optional):

Reviewers' comments:

Reviewer's Responses to Questions

**Comments to the Author**

1. Does the manuscript provide a valid rationale for the proposed study, with clearly identified and justified research questions?

Reviewer #1: Yes

2. Is the protocol technically sound and planned in a manner that will lead to a meaningful outcome and allow testing the stated hypotheses?

Reviewer #1: Yes

3. Is the methodology feasible and described in sufficient detail to allow the work to be replicable?

Reviewer #1: Yes

4. Have the authors described where all data underlying the findings will be made available when the study is complete?

Reviewer #1: Yes

5. Is the manuscript presented in an intelligible fashion and written in standard English?

Reviewer #1: Yes

6. Review Comments to the Author

You may also provide optional suggestions and comments to authors that they might find helpful in planning their study.

Reviewer #1: This appears to be a well thought out and important study to address the needs of users of mental health services that require complex care. The protocol is clear and logical. The only comments that I have are minor.

Page 5 – the 4th study objective, does the investigator mean “who care for a service user” if please amend

Page 9 – It appears that prior work conducted by the investigator has informed this study design. The investigator also states that ethical approval was received on 19th March 2021. Can the investigator please clarify if any data has been gathered directly for this protocol.

7. PLOS authors have the option to publish the peer review history of their article (what does this mean?). If published, this will include your full peer review and any attached files.

Reviewer #1: No

---

## [Editor Report · Acceptance letter]

28 Feb 2022

PONE-D-21-09749 

COMplex mental health PAThways (COMPAT) Study: a mixed methods study to inform an evidence-based service delivery model for people with complex needs: Study protocol 

Dear Dr. Saini:

I'm pleased to inform you that your manuscript has been deemed suitable for publication in PLOS ONE. Congratulations! Your manuscript is now with our production department. 

Kind regards, 

on behalf of

Dr. Kathleen Finlayson 

Academic Editor

PLOS ONE